# Postoperative Hypocalcemia following Non-Cardiac Surgical Procedures in Children with 22q11.2 Deletion Syndrome [note 1]

**DOI:** 10.3390/genes13101905

**Published:** 2022-10-20

**Authors:** Jill M. Arganbright, Meghan Tracy, Max Feldt, Srivats Narayanan, Ashna Mahadev, Janelle Noel-MacDonnell

**Affiliations:** 1Division of Pediatric Otolaryngology, Children’s Mercy Hospital, Kansas City, MO 64108, USA; 2School of Medicine, University of Missouri-Kansas City, Kansas City, MO 64108, USA; 3Division of Pediatric Endocrinology, Children’s Mercy Hospital, Kansas City, MO 64108, USA; 4Health Services and Outcomes Research, Children’s Mercy Hospital, Kansas City, MO 64108, USA

**Keywords:** 22q deletion syndrome, postoperative hypocalcemia, surgical procedures

## Abstract

The guidelines for management of children with 22q11.2 deletion syndrome (22q11DS) highlight the risk for developing hypocalcemia after surgery and recommend monitoring calcium perioperatively. Despite this guidance, little has been published on postoperative hypocalcemia and 22q11DS. Our goals were to evaluate the frequency of perioperative calcium monitoring and examine how often postoperative hypocalcemia was identified. This is a retrospective chart review of patients in our 22q Center’s repository. Inclusion criteria were a diagnosis of 22q11DS and a history of a non-cardiac surgical procedure. Data collected included all non-cardiac surgeries and perioperative calcium labs. In total, 68 patients were included and underwent 305 on-cardiac surgeries. Patients in only 17% of these surgeries had postoperative calcium testing, but of those tested, 58% showed hypocalcemia. Patients with history of hypocalcemia at the time of chart review undergoing non-cardiac surgeries were tested postoperatively 40% of the time; however, 67% of these had hypocalcemia. Similarly, for patients without history of hypocalcemia, postoperative testing occurred 60% of the time, with 52% of these having hypocalcemia. This study demonstrates that postoperative hypocalcemia in children with 22q11DS following non-cardiac surgeries is common and affects patients both with and without prior history of hypocalcemia. These data support establishing a protocol for perioperative testing/management of hypocalcemia for patients with 22q11DS.

## 1. Introduction

22q11.2 deletion syndrome (22q11DS) is the most common microdeletion syndrome and can cause a myriad of health complications, including congenital heart disease, airway and palate anomalies, immunodeficiencies, developmental delay, and endocrinopathies [1]. Hypocalcemia and hypoparathyroidism are common features of 22q11DS. Prior reports have shown that hypocalcemia occurs in 49–80% of patients and up to 69% have abnormal parathyroid gland function [2,3,4,5,6]. Classically these are transient features in the neonatal period that often resolve in the first year of life. However, studies have shown both of these events can occur later in life, particularly during times of biologic stress [7,8].

Current published guidelines for management of children with 22q11DS highlight the risk for hypocalcemia during times of biologic stress, which includes infection, burn, peripartum, and surgery. The guidelines recommend monitoring calcium when these events occur [1]. It is hypothesized that the body’s increased demand for calcium due to the stress of surgery combined with a baseline diminished reserve of parathyroid hormone is the cause of postoperative hypocalcemia [8]. Patients undergoing surgery may also have decreased oral intake perioperatively, which reduces their overall baseline oral intake of calcium. Although surgery is considered a type of biologic stress and a subsequent risk factor for hypocalcemia, little has been published on postoperative hypocalcemia and 22q11DS; the current literature is specific to cardiac surgeries with no published studies for non-cardiac surgeries. Given the paucity of literature on this topic, the aims of this study were to first evaluate how frequently calcium was monitored perioperatively for patients with 22q11DS undergoing non-cardiac surgery at our institutions, and second, when calcium monitoring did occur, how commonly postoperative hypocalcemia occurred. Notably, cardiac surgeries have known differences from non-cardiac surgeries in terms of perioperative calcium levels, including cardiopulmonary bypass which has its own unique risk for developing hypocalcemia separate from 22q11DS. To mitigate the risk of hypocalcemia during cardiac cases requiring bypass, our institution has an existing protocol for perioperative monitoring of calcium for all cardiac surgeries that includes calcium correction throughout the case if needed. Due to the independent risk cardiac bypass poses to patients’ calcium levels, we subdivided the surgical procedures into “cardiac” and “non-cardiac”. For the purposes of this study, we focused on children undergoing only non-cardiac surgeries.

## 2. Methods

### 2.1. Study Design and Participants

We reviewed charts of consecutive patients under the age of 18 years enrolled in our 22q Center’s repository. We included patients with a diagnosis of 22q11DS confirmed by FISH, MLPA, CGH or SNP microarray. Patients were excluded if they did not undergo any surgical procedures at our institution. For each included patient, we collected past medical history including cardiac disease, palatal anomalies, and history of hypocalcemia and/or hypoparathyroidism prior to chart review. All surgical procedures patients underwent at our institution were recorded. Patients were categorized based on the types of surgeries they underwent: cardiac only, non-cardiac only, and mixed (which included patients with history of undergoing both cardiac and non-cardiac surgeries). To limit potential confounding factors, and to create the most homogenous group to aid in analysis, we focused the results of this study on patients who had undergone only non-cardiac surgeries with no prior history of a cardiac surgery. Additionally, for each of the non-cardiac surgical procedures, we reviewed calcium labs pre- and postoperatively. Preoperative labs were considered to be within 6 months prior to surgery, and postoperative labs were considered to be within 48 h. Hypocalcemia was defined as ionized calcium < 1.12 mmol/L or serum calcium < 8.6 mg/dL.

This study was approved by the Institutional Review Board at Children’s Mercy Hospital (IRB #00001921) with a waiver of consent.

### 2.2. Statistical Analysis

The data were descriptively summarized at both the patient and the surgical procedure level. Multiple comparisons were made within each of these data levels for all data collected to assess differences in preoperative versus postoperative testing, presence of hypocalcemia recorded from postoperative testing, history of hypocalcemia prior to surgery, and types of surgeries patients underwent over time. Exploratory analysis was completed to determine which types of surgeries resulted in hypocalcemia and to evaluate time from surgery to first calcium testing. Last, for patients with postoperative hypocalcemia, the trajectory of the calcium recovery was investigated. All statistical analysis was completed using SAS Version 9.4 (SAS Institute, Cary, NC, USA).

## 3. Results

### 3.1. Description of Study Participants

Of the 166 patients reviewed in the 22q repository, 141 had a diagnosis of 22q11DS. Of these 141 patients, 123 had at least one surgical procedure. The mean number of surgeries per patient was 5.0 (range 1–21), resulting in a total of 620 total surgeries. Once patients with mixed-type surgeries and cardiac only surgeries were removed, 68 patients remained that only had non-cardiac surgical procedures recorded at the time of study (Figure 1). The demographic characteristics for our analysis cohort (*n* = 68) were similar to that of all 22q11DS patients from the 22q repository (*n* = 141) (Table 1).

All results reported below include only the most homogenous subset: the 68 patients that had only non-cardiac surgical procedures (*n* = 305 surgeries). Twenty-six (38%) with non-cardiac surgeries were diagnosed by FISH, and 42 (62%) were diagnosed by microarray. The average age at a given surgery was 6.1 years (SD = 4.2 years) with 57% male, and the majority white (85%). Twenty patients (29%) had a history of hypocalcemia noted within their record at the time of chart review, although only 13% (9/68) were taking some form of current supplementation, which included oral calcium, calcitriol, and/or vitamin D at the time of data collection.

### 3.2. Calcium Testing

Two types of postoperative calcium testing were completed: serum calcium and ionized calcium. Calcium testing was reviewed for all patients in the first 48 h postoperatively; calcium levels outside of 48 h were not collected. Serum calcium had a maximum of 6 draws after surgery, while ionized calcium had 10. Serum calcium testing was completed slightly more frequently than testing for ionized calcium. Serum calcium was drawn for 15% of surgeries while ionized calcium was drawn for 7% of surgeries. We found 17/305 surgeries had both types of testing completed postoperatively.

For non-cardiac surgeries, calcium testing was completed preoperatively 50% of the time verses checked postoperatively only 17% of the time. Fourteen percent of these surgeries had both pre- and postoperative testing. For those surgeries in which postoperative testing was conducted, hypocalcemia was found to be present 58% of the time. The average time to calcium testing for serum calcium was 10.7 h (SD = 7.2 h) or for ionized calcium was 14.1 h (SD = 8.2 h) after a given surgery.

### 3.3. Postoperative Calcium Results and Hypocalcemia

Of the surgeries that had postoperative calcium testing, the average serum calcium levels after surgery was 8.3 mg/dL (SD = 3.5 mg/dL), while the average ionized calcium was 1.1 mmol/L (SD = 0.6 mmol/L). Results from postoperative calcium testing found that 28% of the patients (*n* = 18) had hypocalcemia during at least one of their surgeries. Furthermore, we investigated whether a past medical history of hypocalcemia impacted the decision to obtain postoperative testing. Of those with a history of hypocalcemia, 40% had postoperative calcium testing completed of which 67% were diagnosed as being hypocalcemic (Table 2). Conversely, those without a history of hypocalcemia were tested 60% of the time and resulted in hypocalcemia 52% of the time.

Table 3 summarized the data for patients who had postoperative hypocalcemia. When looking at the degree of hypocalcemia for patients undergoing non-cardiac surgeries, we found that the lowest serum calcium value was 5.0 mg/dL. The mean was 8.3 mg/dL, with >75% of the values above 7.8 mg/dL. We also reviewed the types of surgeries (Table 3) at which postoperative hypocalcemia occurred. The most common was pharyngeal flap (*n* = 10), cleft palate repair (*n* = 3), adenotonsillectomy (*n* = 3) and gastrostomy tube (*n* = 3).

For the patients who developed hypocalcemia, we were curious about the overall progression or trend over time. Figure 2 represents all individuals who had postoperative hypocalcemia and more than one recorded serum calcium value. The figure is divided up to show (Panel A) individuals who were monitored with observation alone compared to (Panel B) individuals who were recommended supplementation postoperatively. The results of this figure show no consistent pattern or trend for the calcium levels over the first 48 h with a lot of variability in calcium recovery for both groups.

Finally, despite finding calcium levels as low as 5.0 mg/dL, we had only one patient experience mild symptoms of hypocalcemia (paresthesia), and no patients for whom seizure or other significant associated medical issues were reported.

## 4. Discussion

A summary of data from our cohort of patients with 22q11DS shows that patients undergoing non-cardiac surgeries have a low rate of post-op calcium testing. Interestingly, when testing was completed for patients with non-cardiac surgeries, it was common for them to experience hypocalcemia; this was true for those with a prior history of hypocalcemia and for those with no prior history (Table 2). Notably, while postoperative hypocalcemia occurred relatively commonly, in a majority of these cases calcium levels were considered mildly reduced. These data support the current 22q11DS guideline recommendations for calcium monitoring postoperatively; however there is still uncertainly as to the exact cause of and ultimate clinical impact of the hypocalcemia.

Classically, hypocalcemia and hypoparathyroidism in patients with 22q11DS have been thought to be transient features in the neonatal period. However, studies have shown that both these events can occur later in life, particularly during times of biologic stress. Kapadia et al. found that 8% of their patients had no prior history of hypocalcemia until a documented event occurred at 29 months of age or later [8]. Taylor et al. found that 13–30% of patients have possible hypoparathyroidism outside of the neonatal period and called this laten hypoparathyroidism [7].

The exact etiology for latent hypocalcemia and hypoparathyroidism is not completely understood. One hypothesis by Kapadia et al. is that patients with 22q11DS can have a diminished reserve of parathyroid hormone [8]. When calcium demand goes up due to a biologic stress (e.g., surgery), there is reduced parathyroid hormone availability to respond to this increased need for calcium. The body’s compensatory elevation of parathyroid hormone (PTH) is insufficient; subsequently the calcium supply cannot increase, resulting in a state of hypocalcemia.

Patients undergoing surgery may also have decreased oral intake both pre- and postoperatively, which reduces their overall baseline oral intake of calcium. We should question whether patients undergoing certain types of surgeries, such as tonsillectomy and cleft palate repair, which result in difficult/limited oral intake postoperatively, may experience an even higher risk of postoperative hypocalcemia. Interestingly, a majority of the procedures where patients developed postoperative hypocalcemia did involve the oral cavity/soft palate for which recovery can involve difficult eating/drinking. Some of these surgeries such as adenotonsillectomy and pharyngeal flap have specific diet restrictions in addition to the overall postoperative pain. Further studies are needed to evaluate whether certain types of procedures place children at more risk for developing postoperative hypocalcemia.

Perioperative glucocorticoid (GC) treatment may also exacerbate underlying hypocalcemia and hypoparathyroidism in the postoperative period for patients with 22q11DS. Previous research has suggested that in patients with parathyroid dysfunction, GC treatment may induce clinically significant hypocalcemia [9,10]. GC administration may reduce dietary calcium absorption and may also increase renal calcium excretion. Since patients with 22q11DS may have reduced PTH reserves and lack appropriate PTH responses to hypocalcemia, perioperative GC administration may contribute to postoperative hypocalcemia in this population. Future studies are needed to evaluate the relationship between perioperative GC administration and postoperative hypocalcemia.

Many questions remain, including whether this drop in calcium is clinically relevant or a self-limiting and transient process. Our data suggest that in the majority of cases in which hypocalcemia was present, the hypocalcemia was mildly decreased and resolved within 1–2 days. However, our cohort also had patients with severe hypocalcemia, as low as 5.0 mg/dL, a result that supports testing to potentially prevent such decreases and associated complications (e.g., seizure). Future investigations are also needed to help determine which factors, including type of surgery, length of surgery, and age of patient, may predict higher risk.

Additionally, a better understanding is needed of the sequelae from postoperative hypocalcemia for these patients, including seizures, and also of possible wound healing issues or a more difficult or prolonged recovery. With the lack of literature on this topic for non-cardiac surgeries, we can turn to the literature from our cardiothoracic colleagues that suggests postoperative hypocalcemia does impact surgical outcomes. An article by Shen found that postoperative complications and mortality for congenital heart surgery patients with 22q11DS occurred with greater frequency for those with postoperative hypocalcemia compared to those without postoperative hypocalcemia [11]. Yeoh and colleagues suggested that a low calcium levels in the postsurgical period may play a role in hemodynamic instability or in the development of postoperative seizures and subsequently worsen clinical outcomes for patients with 22q11DS [12]. An article by Yang et al. found higher mortality rates for 22q11DS patients undergoing cardiac surgery, with lower minimum calcium levels and longer duration of hypocalcemia [13]. Cutilio et al. recommends measuring ionized serum calcium both pre- and postoperatively in all patients with 22q11DS particularly in light of the positive correlation between postoperative hypocalcemia and mortality rate [14]. A recent review article by Putotto et al. on conotruncal heart defects in patients with 22q11DS recommends perioperative check of calcium levels for these patients [15]. As stated above, there are several differences between cardiac and non-cardiac cases, particularly with regard to perioperative calcium levels, so it is difficult to correlate the results of these studies with non-cardiac surgeries. Nonetheless, it is interesting to see how outcomes in cardiac surgery are influenced by perioperative hypocalcemia and wonder whether this applies to non-cardiac surgeries as well. At our institution, the lead author’s protocol for perioperative testing for patients with 22q11DS is designed to ensure that the patients have updated calcium labs preoperatively. If the patient is admitted overnight after surgery, a serum calcium will be checked on the morning of postoperative day 1. If the calcium is abnormal, an endocrinology consultation is obtained. Our institution is currently working on developing a formal protocol, and we are aware of other institutions that have already formalized a pathway for perioperative management of calcium for patients with 22q11DS.

Limitations to this study include that this is a retrospective study from a single institution. Currently our institution does not have a set protocol for calcium monitoring in children with 22q11DS; thus, there was a lack of uniformity for testing patients perioperatively which may introduce bias into our current results. The data for the exact time of diagnosis for history of hypocalcemia was not collected and thus we were not able to determine if this diagnosis was present prior to the surgery or potentially given as a result of the surgery or at some point thereafter. Small sample size did not allow us to perform subgroup analysis. Another limitation is that we focused our calcium and lab results to within 48 h after a surgery and did not follow these patients’ values long-term. Additional lab values that could be useful and allow more specific conclusions to be drawn include: magnesium, phosphorus, intact parathyroid hormone, and albumin. We attempted to collect these for this study, however we found they were rarely drawn within the 48-h window following surgery. Ideally a prospective study with a rigorous tracking protocol following patients longitudinally over time would be needed to validate current results and better delineate recommendations.

## 5. Conclusions

This study demonstrates the presence of postoperative hypocalcemia in children with 22q11DS following non-cardiac surgeries. For patients undergoing non-cardiac surgeries, postoperative calcium testing was not common. However, when testing did occur, hypocalcemia was identified frequently. Providers caring for patients with 22q11DS, particularly perioperatively, should be aware that patients with 22q11DS are at increased risk for hypocalcemia following surgery. These data support establishing a protocol for perioperative testing and management of calcium for patients with 22q11DS. Larger prospective studies are needed to continue to investigate the multitude of remaining questions on this topic.

## Figures and Tables

**Figure 1 genes-13-01905-f001:**
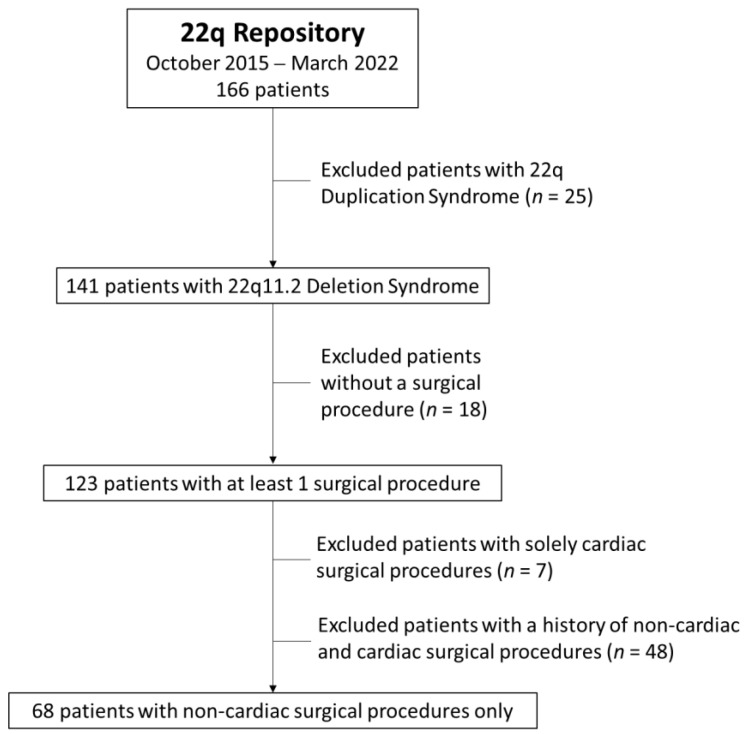
Selection of Study Cohort to obtain homogenous cohort for assessing hypocalcemia.

**Figure 2 genes-13-01905-f002:**
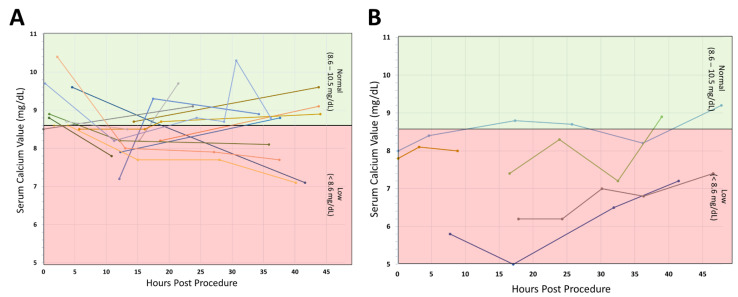
Serum Calcium Levels Recorded within 48 h of Surgery. For individuals who had >1 serum calcium testing recorded, we looked at trajectories of serum calcium levels and if or when levels returned to normal. (**A**) Individuals who were observed conservatively and not recommended any postoperative calcium supplementation. (**B**) Individuals who were recommended some type of postoperative calcium supplementation.

**Table 1 genes-13-01905-t001:** Clinical Characteristics.

	All 22q11DS Patients (*n* = 141)	22q11DS Patients with Only Non-Cardiac Surgeries (*n* = 68)
Age at Surgery (years) mean (SD)	3.6 (4.4)	6.1 (4.2)
Male, *n* (%)	82 (58)	39 (57)
Race or ethnicity, *n* (%)		
White	117 (83)	58 (85)
African American	7 (5)	3 (4)
Asian	1 (1)	1 (1)
American Indian/Alaska Native	1 (1)	1 (1)
Multiracial	4 (3)	1 (1)
Hispanic	9 (6)	4 (6)
Unknown	1 (1)	0 (0)
22q11DS Breakpoints		
A–B	8 (6)	5 (7)
A–C	2 (1)	0 (0)
A–D	54 (38)	30 (44)
B–D	4 (3)	3 (4)
C–D	4 (3)	2 (3)
Other	2 (1)	2 (3)
Unknown	67 (48)	26 (38)

**Table 2 genes-13-01905-t002:** Percentage of postoperative testing and postoperative hypocalcemia with respect to prior history.

	Post-Op Ca * Testing (%)	Post-Op Hypocalcemia (%)
History of Hypocalcemia ^	40	47
No History of Hypocalcemia ^	60	53

^ History of hypocalcemia at the time of chart review. * Ca = calcium.

**Table 3 genes-13-01905-t003:** Postoperative hypocalcemia found in patients who underwent only non-cardiac procedures.

Subject ID	22qDS Breakpoint	Any History of Hypocalcemia	Age at Procedure (Years)	Name of Surgery	Lowest Post-Op Ca ^†^ (mg/dL)	Lowest Post-Op Ionized Ca ^•^ (mmol/L)	Treatment
1	Unknown ^◊^	−	14.6	Mammaplasty	7.5		
1			14.7	Pharyngeal flap	7.3	1.09	
2	A-D	+ *	0.1	Ventriculoperitoneal shunt		1.09	
3	Unknown ^◊^	+ * ^	0.5	Gastrostomy tube	8.2	1.05	+ ^a^
3			1.0	Cleft palate repair	7.9		
3			1.9	ORIF	5	0.73	+ ^ab^
3			15.1	ORIF	6.2	0.82	+ ^a^
4	A-D	−	0.2	Gastrostomy tube	8.2		
4			1.3	Superior adenoidectomy	8.2		
5	A-D	−	3.5	Cleft palate repair	8.5		
5			3.8	Thoracotomy	8.5		
6	A-D	+ *	0.0	Colostomy	7.9		
7	Unknown ^◊^	−	12.8	Spinal Fusion	8.5		
8	A-D	+ ^	10.7	Spinal Fusion	7		
9	A-D	+ ^	6.7	Pharyngeal flap	8.1		
10	Unknown ^◊^	−	12.7	Pharyngeal flap	8	1.09	
11	Unknown ^◊^	+ ^	5.4	Adenotonsillectomy		0.76	
11			5.6	Pharyngeal flap		1.03	
12	A-D	−	6.6	Spinal Fusion	7.8		
12			6.6	Debridement of cervical wound	7.9		
13	Unknown ^◊^	+ *	3.9	Adenotonsillectomy	8.4		
13			9.1	Debridement of scalp wound	8.3		
14	A-D	+ ^	9.0	Pharyngeal flap	7.2	1.05	+ ^a^
15	A-D	+ ^	3.4	Cleft palate repair	8.2		
15			5.3	Adenotonsillectomy	8.4		
15			5.5	Pharyngeal flap	8.4		+ ^a^
16	A-B	+ ^	5.5	Pharyngeal flap	8.2		
17	A-D	−	8.3	Pharyngeal flap	7.8	1.04	+ ^a^
17			11.6	Pharyngeal flap	8		+ ^a^
18	Unknown ^◊^	+ *	5.7	Pharyngeal flap	7.4	0.98	+ ^a^

^◊^ Diagnosis confirmed by FISH testing. + = Positive, − = Negative. * Transient in NICU, ^ Other. ^†^ Normal serum calcium range: 8.6–10.5 mg/dL. ^•^ Normal ionized calcium range: 1.13–1.37 mmol/L. a = oral calcium, b = oral Calcitriol/Rocaltrol.

## Data Availability

The data presented in this study are available on request from the corresponding author. The data are not publicly available due to privacy.

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
