# Peer review of "Postoperative Hypocalcemia following Non-Cardiac Surgical Procedures in Children with 22q11.2 Deletion Syndrome†"

_genes, 2022, doi:10.3390/genes13101905_

Round 1

Reviewer 1 Report

In this article, the authors report on a study on postoperative hypocalcemia in a sample (n=73) of children with 22q11.2 deletion syndrome with a history of non-cardiac surgical procedures. Given that, as the authors discuss, little has been published on postoperative hypocalcemia in 22q11.2DS, and the risk for hypocalcemia during times of stress such as surgery, this kind of studies is much needed to inform evidence-based clinical practice guidelines. As such, this study relates to an important issue for study.

In my view however, the manuscript would benefit from some restructuring and clarifications, as well as more consistency in reporting. Also, for the sake of consistency, I would suggest the authors to discuss only about the patients who underwent only non-cardiac surgeries. For example, in the title it is suggested that only non-cardiac surgeries are studies, in the goal (abstract) cardiac surgeries are not excluded, in the methods it is mentioned that the authors ‘primarily’ focused on non-cardiac surgeries, etcetera.

Some specific comments and suggestions:

Title.

-              I suggest to change the word ‘Patients’ to ‘Children’ to clarify that only children were studied.

Abstract.

-              It would be helpful for the reader if the authors can clarify if ‘history of hypocalcemia’ was prior to the surgery, or at the time of the chart review.

Methods.

-              2.1 ‘Confirmed diagnosis if 22q11DS’. Can the authors please clarify what method were used to confirm the diagnosis. Molecularly confirmed? All patients had a 22q11.2 deletion incorporating the LCR22A-LCR22B region?

-              2.1 ‘prior hypocalcemia and/or hypoparathyroidism’. Prior to? The surgery or to the chart review?

-              2.1 Line 69-75 consider moving the background information on cardiac surgeries vs non-cardiac surgeries up to the Introduction.

-              2.2 Mention is made about statistically analysis in the Methods, but I don’t see any results of statistical analysis in the Results. (Just proportions/percentages are mentioned in the results).

Results.

-              The authors may want to consider adding a Table summarizing patient characteristics and/or study results.

-              The authors will want to make a decision if they report on the patients with history of cardiac surgeries, or not.

-              It would be helpful to see the sex distribution.

-              Consistency in reporting, e.g., NUMBER (PERCENTAGE with one decimal). Now, alternately the number of patients, the percentage, or both are provided, with one or no decimals for the percentages.

-              3.1 The first sentence is redundant. This information is already provided in the Methods section.

-              3.1 ‘and/or vitamin’. Please clarify if this is vitamin D.

-              3.1 Is there something to say about the non-cardiac surgeries? What kind of surgeries should we think of?

-              3.2 Is there something to say about how many children received treatment for hypocalcemia (including vitamin D supplementation) pre- and postoperatively?

-              3.3 I think that it would be insightful for the reader to see the individual calcium results in a plot. Now, it is difficult to get an idea how serious these hypocalcemia levels were, and distribution of the values, especially for those with values lower than 7.8 mg/dL.

-              3.3 Line 125-127 Please move to the Discussion.

-              Line 141-158 I would suggest to move this up to earlier in the Results section.

Discussion.

-              I would suggest the authors to start outlining their own results and placing them in the context of what is already known.

-              The limitation section needs more elaboration. For example, regarding the effect of bias, and the impact of the absence of systematic evaluations and a comparison group.

Reviewer 2 Report

The manuscript presented for review, describing a retrospective assessment of the calcium level in the body of patients with 22q11.2 microdeletion after exposure to the stress factor at this point it was surgery, is interesting and may have a significant impact on the creation of e.g. diagnostic recommendations of calcium measurments before/during and after surgery in patients with this disease, especially in the context of severe complications related to lowering / low levels of calcium. However, before the presented work is published, it requires changes and corrections:

1-    in the section Study design and participants, there should be a sentence summarizing how many patients were actually included in this study, additionally there should be information about the age range of these patients (mean / median standard deviation) and information about the gender of the patients (no. of male/female). Perhaps creating an additional scheme for including / creating a study group would organize this description. In how many patients in the study group were simultaneously assessed for calcium levels before, during and after surgery? Was total calcium only or also ionized calcium assessed ? An additional table would summarized this information

2-      in the Statistical analysis section, the authors inform about the use of the Chi ^ 2 test, although later in the description of the results, p values for statistical calculations are not given. A table showing the exact division into subgroups, N values (%) and p values for statistical analysis could be useful. Which p-value was considered statistically significant?

3-      in the Description of Study Participants section - the authors provide information about the number of samples taken for the assessment of total calcium and ionized calcium at what time intervals did these individual samples were taken? Did patients receive medications or intravenous infusions between these blood drews that could affect calcium levels?

4-     page 3, line 115-116 - incomprehensible sentence - are they the same patients?

5-     page 3, line 121 - what were the calcium concentration ranges in this group? whether it was total or ionized calcium

6-     after retrospective analysis, were the patients divided according into total / ionized calcium concentration ranges and is statistical analysis between these subgroups were done? Was such an statistical analysis between subgroups of patients according to:  on age / gender or the number of surgical procedures performed? (Using of table would summirized this kind of information (with p-value))

7-     there any complications resulting from calcium reduction in the subgroups of patients with different calcium levels / what kind of complications/ was statistical analysis performed in these subgroups?

8-      in the case of figure 1, does the information regarding calcium supplementation refer to its supplementation before or after surgery?

Additionally,

Within the study group, it would be useful to divide patients according to information about the calcium concentration of these patients before, during and after surgery (e.g. Table with Mean Concentrations / Medians); There is also no information on the values ​​of PTH, vitamin D or kidney function parameters in these patients, which may also affect the level of calcium in the body. In addition, the test group should be divided according to whether total or ionized calcium or both were tested; Because the value of ionized calcium is the most important cause it is a biologically active form. All this information would allow the authors to draw more specific conclusions or even propose the best diagnostic procedure for this disease entity
